# Spatial-Temporal Evolution and Its Influencing Factors on Urban Land Use Efficiency in China's Yangtze River Economic Belt

Liguo Zhang [1], Luchen Huang [1,2], Jinglin Xia [1] and Kaifeng Duan [3,*]

1   School of Economics, Jiangxi University of Finance and Economics, Nanchang 330013, China
2   Department of Mathematics and Statistics, Chonnam National University, Gwangju 61186, Republic of Korea
3   School of Economics and Management, Fuzhou University, Fuzhou 350108, China
*   Correspondence: kefee920729@fzu.edu.cn; Tel.: +86-180-0562-1443

**Abstract:** Improving urban land use efficiency is a feasible way to realize sustainable development and alleviate urban land pressure on the city. The main purpose of this article is to measure the urban land use efficiency of the Yangtze River Economic Belt, and explore its evolutionary trends and influencing factors, so as to provide references for policy formulation to promote efficient land use and sustainable development. Therefore, we calculated the value of urban land use efficiency in the Yangtze River economic belt from 2004 to 2019, based on the super efficiency SBM model, including unexpected output. Further, we analyzed the spatial-temporal evolution, and spatial correlation and its influencing factors. The main results are as follows: Firstly, urban land use efficiency in the Yangtze River economic belt continues to improve as a whole, but it is higher in the east and lower in the west. In the kernel density evolution map, the development trend is steep at first and then slows, and the gap tends to decrease. Secondly, the spatial correlation of urban land use efficiency in the Yangtze River economic belt increases year by year, showing a positive correlation overall. The high-high agglomeration shifts to the east, low-low agglomeration shifts to the west, and low-high and high-low agglomeration show scattered distribution. The hot and cold spots are distributed regionally and have a diffusion trend. Thirdly, the results of the spatial Dubbin model show that the urbanization level, government expenditure and industrial instruction transformation can promote the improvement of urban land use efficiency, and people density and land use scale can inhibit its improvement. Additionally, there is remarkable heterogeneity in the effect of these influencing factors. On the whole, the effect of non-resource-based cities is better, and it is more so in the cities of the eastern region.

**Keywords:** urban land use efficiency; Yangtze River Economic Belt; temporal and spatial evolution; Un_Super_SBM model

## 1. Introduction

As a major strategic development area in China, the Yangtze River Economic Belt (YREB) has shown a strong development trend in recent years. Although it covers 21.5% of the country's land, it provides a platform for economic activities for more than 42% of the population and contributed more than 46% of China's GDP in 2021 [1]. However, with the promotion of urban-rural integration, the urban population is rapidly increasing, and urban construction land continues to occupy cultivated land. We are facing problems of extensive development and low efficiency, which calls for an urgent solution for urban low land efficiency [2,3]. In 2018, General Secretary Xi Jinping pointed out at the Symposium of Further Developing the YREB that, under the new situation, we must adhere to the new development thinking, focus on large-scale protection instead of large-scale exploration, and cooperate with regions to jointly promote high-quality development of the YREB. The fifth Plenary Session of the 19th Central Committee proposed effectively utilizing land resources and promoting the new pattern of urbanization with people at the core. Urban

land use efficiency (ULUE) is calculated by the input land resources and unexpected output, which establishes a relationship between urban operation and land utilization. Researching improvements in ULUE is an effective entry point to promote coordinated and sustainable regional development and relieve urban land pressure [4,5].

How to evaluate the ULUE? Over the past years, a large interest has developed in the calculation and evaluation of ULUE. At present, the research on the evaluation of ULUE mainly focuses on the necessity of ULUE measurement, improvement of measurement methods, and promotion strategies of ULUE. First, most scholars believe that it is necessary to measure ULUE. For example, Nesru et al., through the analysis of ULUE in Ethiopia, found that low efficiency is a common phenomenon, and sustainable urban development needs to improve ULUE [6]. Lejandro et al., through the analysis of ULUE in Mexican cities, found that low efficiency is accompanied by high pollution [7]. Yang et al. analyzed the land-use situation of 115 cities in China and believed that resource accumulation and economic structure are especially important for the improvement of ULUE [8]. Second, there are significant differences in the methods of ULUE measurement applied in different research. On the one hand, in terms of index system, there are large differences in the selection of different scholars. Some scholars believe that the principle of benefit maximization should be followed in the process of land use, so it is better to directly reflect the land output through economic indicators [9,10]. However, there exist different conclusions as to whether ecological environment constraints need to be considered in the process of land use, and pollution in the production process should be considered for measuring ULUE [11,12]. On the other hand, in terms of model construction, efficiency measurement approaches include the parametric, non-parametric, multi-index dimensionality reduction, and other methods. Some scholars measure land efficiency by constructing the DEA-Malmquist index method [10,13]. While other scholars analyze it by constructing production functions, such as the Stochastic Frontier Approach (SFA) [14–16]. With further research, many scholars recognized that traditional DEA models, such as BBC or CCR, either require proportional change on input and output or constant input or output based on the same assumption, which is a large deviation. When there is insufficient output or excessive input, the traditional DEA model will overestimate the efficiency value, which is inconsistent with reality [17]. Tone proposed a new efficiency measurement named SBM, which is based on slack variables to solve the possible errors of the DEA model [18]. Yang et al. used the data of the Yangtze River Delta region and based on this model evaluated the value of ULUE. They found that this method can make up for the shortcomings in measurement methods, which further confirmed the research of Tone [19]. Third, there are multiple perspectives on the promotion strategy for ULUE. On the one hand, there are differences in the methods of evaluating ULUE changes. Some researchers have evaluated the differences and changes in ULUE from the aspects of the decomposition effect coefficient [20] and slack variable [21]. Other research evaluated the spatial-temporal evolution of ULUE by the kernel density method [22], spatial distribution by the spatial-temporal distribution map [9,23], and spatial relevance by the hot and cold spot distribution map [16,24]. On the other hand, some scholars have analyzed the influencing factors of ULUE by constructing econometric models and researched them from the perspectives of industrial structure [25], population [26], and urbanization [27], respectively.

To date, there has been much research on the evaluation of ULUE, and it has provided more guidance for follow-up research. However, from the current study, further research is still needed from the following two aspects: First, the existing literature is mostly evaluated from the ULUE of provincial cities or single prefecture level cities, and there are few studies involving the overall and local city in the YREB. Second, most of the existing models for measuring ULUE consider the unexpected output, but the efficiency value measured by the SBM model can only be between 0 and 1, which cannot analyze the effective decision-making unit, and it needs to be further improved.

Based on the previous research on ULUE, the marginal contribution of this article has two points: First, the super efficiency SBM model based on the unexpected output

value (Un_Super_SBM) is used to measure the ULUE, which is improved on the basis of the traditional measurement of ULUE. Secondly, the changes in and influencing factors on ULUE in the YREB are discussed in depth, and the changes in ULUE are analyzed in detail through time series, spatial autocorrelation, spatial Dubin regression, and the heterogeneity test. We have three specific purposes for this article: 1. Evaluate ULUE based on the panel data of 110 cities in the YREB from 2004 to 2019, and we use the Un_Super_SBM model to calculate ULUE. 2. Explore the trend of spatial-temporal evolution. Based on the calculation, we use the spatial distribution map, kernel density function, Lisa map, and cold and hot spot map to describe the spatial-temporal evolution and autocorrelation of ULUE. 3. Analyze the influencing factors of ULUE. By constructing the SDM model, we will explore the influencing factors and test whether there is a spatial spillover effect of urbanization on ULUE. Further, we divide the cities in the YREB by resource-type attribute and location to investigate the heterogeneity characteristics of ULUE. We aim to contribute our results and proposals for empirical reference and theoretical support for the formulation of land-related policies in the urbanization process in various regions.

This article is composed of four sections. The organization is as follows: Section 1 is the introduction, and it mainly considers the research background and significance. Section 2 is the materials and methods, and it considers the study area, research methods, and data. Chapter 3 analyzes the data gathered and presents the results and, in turn, solves the problems discussed. Chapter 4 is the discussion and conclusions, and discusses the results and presents the conclusions of the research.

## 2. Materials and Methods

### 2.1. Research Area

In 2016, "The Outline of the Development Plan of the Yangtze River Economic Belt" was proposed, which aimed to establish a new development pattern. This policy emphasizes the spatial layout and functional orientation of the YREB, and proposes to promote new urbanization and build a new pattern of East-West two-way and land-sea overall planning [28,29]. Under the guidance of this policy, the YREB has received further development support. The YREB across China, which contains western, central, and eastern regions, links nine provinces and two cities, including Shanghai, Jiangsu, Hunan, Anhui, Zhejiang, Hunan, Jiangxi, Sichuan, Hubei, Chongqing, Guizhou, and Yunnan (Figure 1).

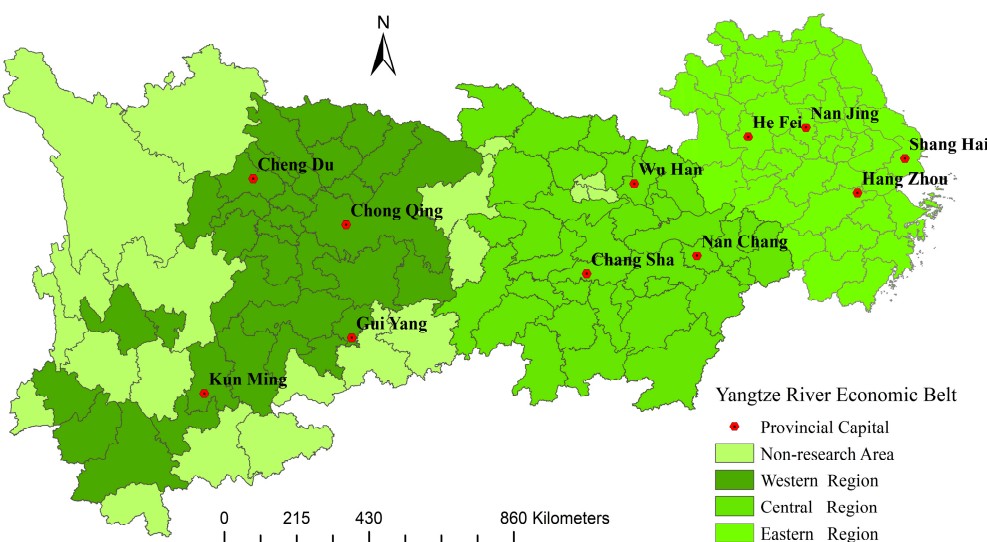

**Figure 1.** Regional map of the Yangtze river economic belt.

The YREB covers an area of 2,052,300 square kilometers, about 21.5% of China [30,31]. With high industrial agglomeration and a tight land area, the YREB has a high demand for land resources for regional development. In recent years, rough and crazy development has

significantly reduced ULUE. At the same time, due to the large regional span, the economic development in the YREB presents a pattern of "high in the east but low in the west" [32]. According to the data of the statistical yearbook, the GDP of YREB in 2020 will exceed 47 trillion-yuan, accounting for more than 46% of China's. However, there is a substantial difference between different regions. The GDP of the eastern region is 24.5 trillion-yuan, central region 11.1 trillion-yuan, and western region 11.6 trillion yuan. Because of these characteristics of YREB, the region provides a very typical case for exploring the coordinated development, temporal and spatial characteristics, and development of regional urban land. Although YREB's regional economic development is far ahead, it also faces many problems. According to the data of "The Ecological Development Report of the Yangtze River Economic Belt (2019–2020)", the water and soil loss has continued in recent years. In 2019, the area of water and soil loss will reach 293,900 square kilometers, accounting for 20.14% of the land area. Although the pollutants and energy consumption of YREB will decrease in 2020, the situation is still not optimistic. In 2020, YREB's total wastewater discharge will account for 44.4% of the country's total, and the proportion of wastewater will exceed 40%. Therefore, efficient and green land use is of great significance to sustainable economic development, and more attention should be paid to urban land with concentrated population distribution. By analyzing this ULUE, it can provide a reference for urban land use planning in China and other major economic integrated regions in the world.

### 2.2. Research Methods

### 2.2.1. Un_Super_SBM Model

Tone (2002) proposed a model based on undesired output and slack variables (Un_SBM) based on the SBM model [33]. It assumes that we have I decision-making units (DMUs), where the input vector is $X = \left( x_{ij} \right) \in R^{m*n}$, the output vector is $Y^y = \left( y_{kj} \right) \in R^{s1*n}$ and the undesired output is $Y^z = \left( y_{kj} \right) \in R^{s2*n}$. When we consider that $X$, $Y^y$ and $Y^z$ are all greater than zero, we can get the possible production set as follows:

$$P = \{(x,y)|x \geq X\Lambda, y^y \leq Y^y\Lambda, y^z \geq Y^z\Lambda, \Lambda \geq 0\} \tag{1}$$

where, $\Lambda = [\lambda_1, \lambda_2, \cdots \lambda_n] \in R^n$ indicates the weight coefficient; $x \geq X\Lambda$ indicates that the actual input level is greater than the frontier input level; $y^y \leq Y^y\Lambda$ indicates that that the actual output level is less than the frontier output; $y^z \geq Y^z\Lambda$ indicates that the unexpected actual output level is greater than the frontier output [34].

According to the condition, we can calculate the $DMU$ $(x_0, y_0, z_0)$ by the Un_SBM model. The equation is as follows:

$$
\begin{aligned}
min\ \rho &= \frac{1 - \frac{1}{m}\sum_{i=1}^{m}\frac{s_x^i}{x_{i0}}}{1 + \frac{1}{s1+s2}\left( \frac{1}{s}\sum_{k=1}^{s1}\frac{s_k^y}{y_{k0}} + \frac{1}{s}\sum_{l=1}^{s2}\frac{s_l^z}{z_{l0}} \right)} \\
s.t.\ &x_{i0} = \sum_{j=1}^{n}\lambda_j x_j + s_i^x, \forall i; \\
&y_{k0} = \sum_{j=1}^{n}\lambda_j y_j - s_k^y, \forall k; \\
&z_{l0} = \sum_{j=1,\neq 0}^{n}\lambda_j z_j - s_l^z, \forall l; \\
&s_i^x \geq 0,\ s_k^y \geq 0,\ s_l^z \geq 0, \lambda_j \geq 0, \forall i, j, k, l.
\end{aligned}
\tag{2}
$$

where, $s_i^x \in R^m$, $s_k^y \in R^{s1}$, $s_l^z \in R^{s2}$ represent too much input, too little expected output and too much undesired output, respectively. When $\rho$ = 1, DMU is efficient and it's invalid when $\rho < 1$.

Because the value of efficiency based on the Un_SBM model can only be between 0 and 1, a comparison between efficient DMUs cannot be achieved. To further improve this

model, we refer to the research of Chen for analysis [35]. Therefore, based on the Un_SBM model, we can construct the Un_Super_SBM model, and the equation is as follows:

$$
\begin{aligned}
min\ \rho &= \frac{1 + \frac{1}{m} \sum_{i=1}^{m} \frac{s_x^i}{x_{i0}}}{1 - \frac{1}{s1+s2} \left( \frac{1}{s} \sum_{k=1}^{s1} \frac{s_k^y}{y_{k0}} + \frac{1}{s} \sum_{l=1}^{s2} \frac{s_l^z}{z_{l0}} \right)} \\
s.t.\ & x_{i0} \geq \sum_{j=1}^{n} \lambda_j x_j + s_i^x, \forall i; \\
& y_{k0} \leq \sum_{j=1}^{n} \lambda_j y_j - s_k^y, \forall k; \\
& z_{l0} \geq \sum_{j=1,\neq 0}^{n} \lambda_j z_j - s_l^z, \forall l; \\
& 1 - \frac{1}{s1+s2} \left( \frac{1}{s} \sum_{k=1}^{s1} \frac{s_k^y}{y_{k0}} + \frac{1}{s} \sum_{l=1}^{s2} \frac{s_l^z}{z_{l0}} \right) > 0 \\
& s_i^x \geq 0,\ s_k^y \geq 0,\ s_l^z \geq 0, \lambda_j \geq 0, \forall i, j, k, l.
\end{aligned}
\tag{3}
$$

where, $s_i^x$, $s_k^y$, $s_l^z$ represent the result of super-efficient ($\rho > 1$). However, we need to note that these variables are not slack variables in the Un_SBM model, and they only represent the part of $\rho > 1$. So, we should combine the result of the Un_SBM model and the result of the Un_Super_SBM model to calculate the final efficiency value.

2.2.2. Kernel Density Estimation

Kernel density estimation (KDE) is a nonparametric estimation. KDE makes no assumptions about the population distribution and is based on randomly drawn samples for the study of characteristics and distributions [36]. The equation is as follows:

$$
\hat{f}(x, h) = \frac{1}{n} \sum_{i=1}^{N} K_h(x - x_i)
\tag{4}
$$

where, $\hat{f}(x, h)$ represents the probability density function of $x$, $h$ represents bandwidth, and $N$ represents the number of observations. $K_h(\cdot)$ is a kernel function through the Gaussian, Tophat, Epanechnikov, and other kernel types. We use the method referred to in Burkhauser et al. to plot the kernel density function map of ULUE in the YREB. By observing its current characteristics, such as its peak value and location, the evolution characteristics of ULUE can be estimated [37]. Specifically, the right shift of the curve means that the efficiency is gradually improving; the reduction in the number of peaks means that the efficiency value is gradually converging; and the shorter the left and right trailing lengths are, the shorter the gap between regions is narrowing. In this article, we take the Gaussian kernel function and set the bandwidth to 1 to estimate ULUE. We can see the function as follows:

$$
K(x) = \frac{1}{\sqrt{2\pi}} \exp\left( -\frac{x^2}{2} \right)
\tag{5}
$$

2.2.3. Spatial Correlation Analysis Model

The correlation between factors increases with geographical proximity [38]. Because of the flow of factors and convenient transportation between cities, there may be spatial correlations in ULUE, so it can be analyzed whether it is geographically related. We construct a spatial adjacency weight matrix in our study and test it with the global Moran index (Moran's I). The formula for calculating Moran's I is as follows:

$$
I = \frac{n \sum_{i=1}^{n} \sum_{j=1}^{n} W_{ij}(x_i - \overline{x})(y_j - \overline{y})}{\sum_{i=1}^{n} \sum_{j=1}^{n} W_{ij}(x_i - \overline{x})^2}
\tag{6}
$$

where, $x_i$ and $y_j$ represent the ULUE of region $i$ and region $j$, respectively, and $W_{ij}$ represents the spatial adjacency weight matrix. In this formula, if the result of I is significant, it indicates that there is a spatial correlation in the whole. And if I is significantly positive, it indicates a positive spatial correlation as a whole; otherwise, it is a negative correlation.

Based on the global Moran index, we further analyze the local spatial correlation. Specifically, we analyze the ULUE of YERB by calculating the LISA index and hot spot and cold values. The formula of LISA and the $G_i^*$ index is as follows:

$$I_i = \frac{(x_i - \overline{x})}{\sum_{i=1}^{n}(x_i - \overline{x})^2} \sum_{j=1}^{n} W_{ij}(x_i - \overline{x}) \tag{7}$$

$$G_i^* = \sum_{j=1}^{n} W_{ij}(d)X_j / \sum_{j=1}^{n} X_j \tag{8}$$

where, $I_i$ represents the Lisa index, and $G_i^*$ represents the cold and hot spot values. When $I_i$ is significant, it shows correlation in the local area. When $G_i^*$ is significant, it will reflect the correlation strength in the local area.

### 2.2.4. Econometric Model Construction

Based on the above mechanisms and research assumptions, we first construct an OLS double fixed model to analyze the influence of ULUE. The formula is as follows:

$$ULUE_{i,t} = \alpha_0 + \alpha_1 X_{i,t} + \delta_i + \nu_t + \varepsilon_{i,t} \tag{9}$$

where, ULUE is the dependent variable, which is employed to measure the ULUE, and $X_{i,t}$ is the independent variable. $\alpha_0 \sim \alpha_1$ represents the coefficients to be estimated, $\delta_i$ and $\nu_t$ are the fixed time and region, respectively, and $\varepsilon_{i,t}$ is the random disturbance term.

Based on the OLS double fixed model, we construct a spatial adjacency weight matrix to test the significance. When Moran's I is significant, we will construct a spatial panel model for the regression analysis based on Formula (10). The formula is as follows:

$$\begin{aligned} ULUE_{i,t} = \chi_0 + \rho W_{i,t}ULUE_{i,t} + \chi_1 Urban_{i,t} + \chi_2 W_{i,t}Urban_{i,t} \\ + \alpha_3 X_{i,t} + \chi_4 W_{i,t}X_{i,t} + \delta_i + \nu_t + \varepsilon_{i,t} \end{aligned} \tag{10}$$

where, ρ is the spatial lag coefficient of the dependent variable, $\chi_1 \sim \chi_2$ is the regression coefficient and the spatial lag coefficient of the independent variable, respectively; and $\varepsilon_{i,t}$ is the random error. When ρ is significant, we can analyze the spatial effect and it is greater than 0 for a positive correlation and less than 0 for a negative correlation. However, since there are many models, the specific choice needs to be further examined.

### 2.3. Research Variable and Data Source

Based on the existing literature and actual situation of YREB, the input indicators of this article are land, capital, and labor force; the desirable output indicator is the added value of the secondary and tertiary industries; and the undesirable output indicators are industrial waste gas and wastewater discharge. The selections of values are shown in Table 1 [14,19,36–40].

**Table 1.** Quantitative indicators of ULUE.

|  | Index | Variable | Unit |
|---|---|---|---|
|  | Land | Built-up area | Km$^2$ |
| Input | Capital | Total fixed-asset investment | million |
|  | Labor | The number of employments in the secondary and tertiary industries | 10,000 people |
| Expected output | Economic Effect | The added value of the secondary and tertiary industries | million |
| Undesired output | Industrial "Three Wastes" | Industrial wastewater discharge | 10,000 t |
|  |  | Sulfur dioxide emissions | t |
|  |  | Industrial soot emissions | t |

The dependent variable is ULUE, which is the result calculated by the the Un_Super_SBM model. Regarding the independent variables, we chose from six aspects: urbanization level (Urban), population density (PD), government expenditure (Gov), industrial structure

transformation (IST), and land use scale (LUS). Scholars have adopted different standards to measure the extent of urbanization. There were mainly two types of measures of urbanization in the past: population and land. However, this study found that these two measurement indicators are relatively simple and cannot reflect the real level of urbanization. Therefore, we refer to the research of Zheng et al. and select the night-light index as a measure of the urbanization level [41]. The night-light index obtains the illumination of the region through remote sensing data, which can avoid human intervention and objectively reflect the urbanization level of the region. At present, the night-light data mainly includes two sets of data: DMSP_OLS and VIIRS_VNL, but DMSP is only updated to 2013. Therefore, we refer to the research of Elvidge to fit the corrected data of the two sets of data and finally obtain the results of the urbanization level in the YREB [42,43]. The indicators of PD, Gov. IST, and LUS are expressed by the population per square kilometer, proportion of fiscal budget expenditure, ratio of added value of tertiary industry to secondary industry, and ratio of construction land area. The calculation equation is shown in Table 2.

**Table 2.** Model variable selection.

| Influence Factors | Variable | Measure |
|---|---|---|
| Dependent Variable | ULUE | Results of Un_Super_SBM |
| Independent Variable | Urban | Night-light data by fitting the corrected data of DMSP_OLS and VIIRS_VNL |
| | PD | Total population/Administrative area |
| | Gov | Fiscal Budget Expenditure/GDP |
| | IST | Added value of tertiary industry / Added value of secondary industry |
| | LUS | Construction land area / Administrative area |

The night-light data comes from the NOAA-funded EOG website: https://eogdata.mines.edu/products/vnl/ (accessed on 29 November 2022). The data of PD, Gov, IST, LUS comes from China City Database, China Urban Statistical Yearbook, and China Statistical Yearbook. For some missing data, the interpolation method and exponential smoothing method were applied for imputation; and for severely missing and discontinuous areas, deletion processing is adopted. The results data were integrated to form panel data about 110 cities in the YREB from 2004 to 2019.

## 3. Results

### 3.1. Time Series Characteristics and Spatial-Temporal Evolution Analysis of ULUE

3.1.1. Time Series Characteristics

According to the assumptions of Formulas (2) and (3), based on the MATLAB R2020 platform and the Un_Super_SBM model, we calculated the ULUE of 110 cities in the YREB from 2004 to 2019. At the same time, to observe the trend of time series characteristics of ULUE, we plotted the average value of the YREB in three regions from 2004 to 2019 based on the Stata16.0 software platform. (Figure 2).

The results show that the ULUE was in a fluctuating upward trend, and the upward trend was the fastest after 2013, with the highest value being 0.562. The evolution trend of sub-regions shows that the temporal characteristics of the eastern, central, and western regions show obvious differences. The average ULUE in the eastern region is the highest, always above the average of the YREB, with the highest value of 0.649. The central region and western region showed cross changes; that is, the average ULUE in the western region was higher before 2013 and higher in the central region after 2013, and exceeded the average value of the YREB. The highest value in the central region is 0.569, and the highest value in the western region is 0.454. The figure shows that the average ULUE is low in the YREB as a whole or sub-regions. From the time series evolution results of ULUE in the YREB and its three regions, it can be found that the overall level of ULUE is low, especially in the central and western regions. This is because we not only consider the effect of economic

development, but also consider the pollution emissions in the production process when measuring ULUE. In the past few years, the YREB has gathered heavy industry and heavy pollution industry. Two of the four major industrial provinces are located in the western region and one in the central region. The distribution of high energy consuming industries and high pollution industries has a trend of "decreasing from east to west", so the ULUE is high in the east and low in the middle and west. We can find that since the implementation of the major strategy of the YREB in 2016, ULUE has been significantly improved, which shows that the policy implementation has received a good response. Based on the time series trend, we plotted the spatial-temporal distribution in 2004, 2009, 2014, and 2019 through the ArcGIS 10.7 software platform to observe the spatial-temporal evolution of ULUE (Figure 3) [17,41].

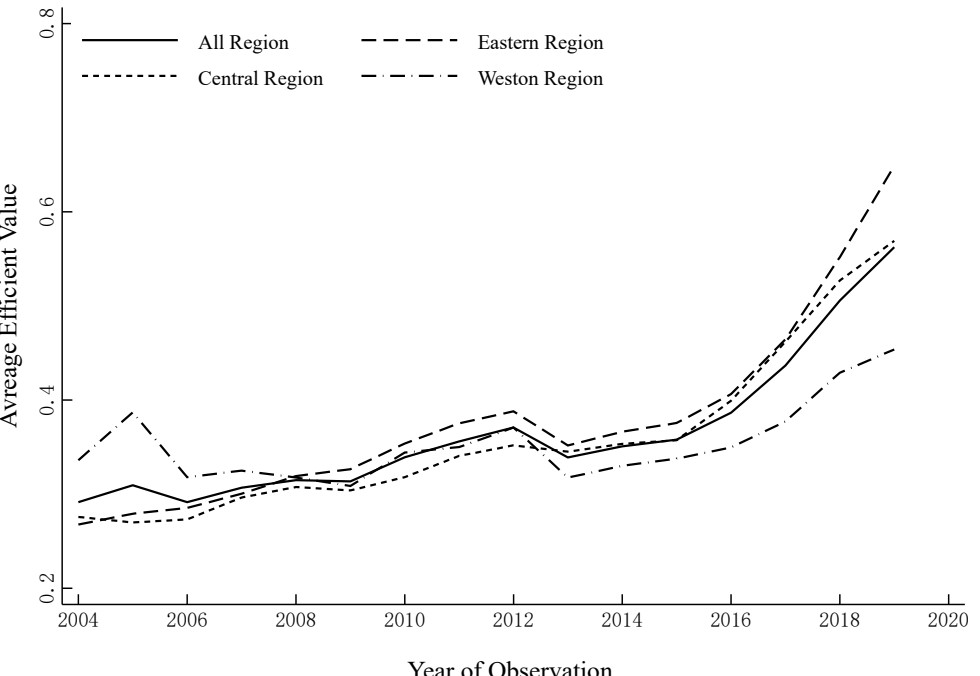

**Figure 2.** Time series trend of ULUE.

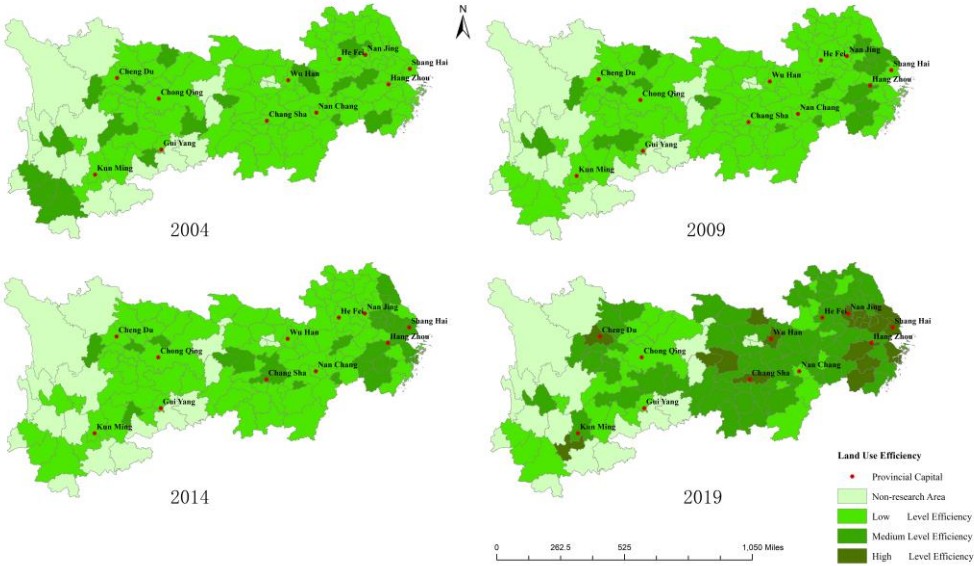

**Figure 3.** Spatial-temporal distribution of ULUE.

Figure 3 shows the distribution results of ULUE in 110 cities of the YREB in 2004, 2009, 2014 and 2019. Overall, the ULUE of the YREB has obvious regional differences and changes significantly over time. In 2004, 17 cities were in the interval of mid-efficiency, and 93 cities were in the relatively low-efficiency interval. In 2009, 20 cities were in the mid-efficiency interval, and 90 cities were in the relatively low-efficiency interval. In 2014, 25 cities were in the medium-efficiency interval, and 85 cities were in the low-efficiency interval. In 2019, 20 cities were in the relatively high-efficiency interval, 61 cities were in the mid-efficiency interval, and 29 cities were still in the low-efficiency interval. It can be found that, as time progresses, these areas of low ULUE are gradually decreasing, and moderate and high ULUE are gradually increasing. The evolution characteristics of temporal and spatial distribution clearly show the trend of temporal and spatial changes. Since the introduction of the planning policy of the YREB, the ULUE has been significantly improved, which is consistent with the result shown in Figure 2. The ULUE has changed significantly from 2014 to 2019. In 2019, high-efficiency regions were concentrated in the eastern cities, middle efficiency in the central cities, and low efficiency in the western cities. Although the efficiency has been improved, the overall efficiency is not particularly high, because it is difficult to recover the resource consumption and ecological damage caused by the rapid economic development in the past in a short time. Additionally, because the transformation of industrial equipment and production modes requires a long period, the overall ULUE is not high, and it is necessary to gradually promote the improvement of ULUE.

### 3.1.2. Spatial-Temporal Evolution Analysis

To further investigate the dynamic evolution trend of ULUE, based on the Stata16.0 software platform, we used the data of 2004, 2009, 2014, and 2019 to estimate the kernel density and draw a kernel density map (Figure 4).

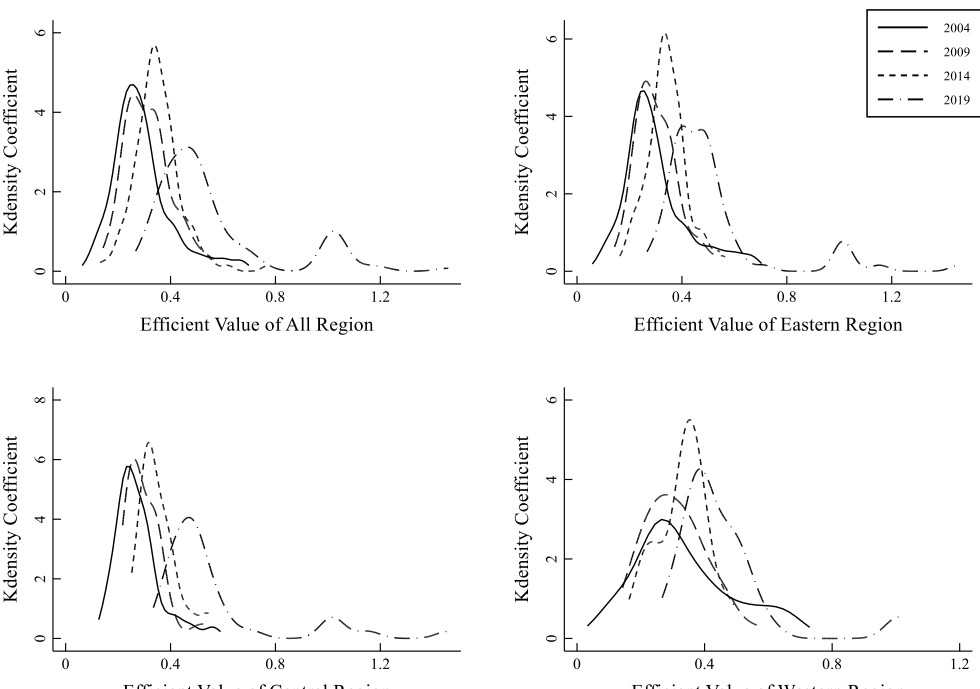

**Figure 4.** Kernel density map of ULUE.

Figure 4 shows the evolution of the KDE in the overall and local regions of the YREB. From the perspective of shape, the overall results of ULUE showed a single peak distribution of "N-type" in 2004, 2009 and 2014 and a double peak distribution of "M-type" in 2019. The wave crest experienced the change of "steep first and then slow," indicating that the ULUE experienced the change in "differentiation first and then narrowing". According

to results of the position, the overall KDE has a tailing phenomenon after 2014. The transition is a single peak first and double peak last, which indicates that the ULUE is gradually increasing, and there is a phenomenon of agglomeration in the high-efficiency range during this period. From the KDE results of the local area, it can be found that the dynamic evolution of ULUE is quite different. From the perspective of shape, the eastern and central regions showed an "M-type" double-peak feature in 2019; the remaining years showed an "N-type" single-peak distribution; and the western region always showed an "And-type" single-peak distribution. According to results on the position, there is a tailing phenomenon in the eastern, central, and western cities, and the tailing in the central and eastern cities is longer and the efficiency value exceeds 1, indicating that the ULUE in the eastern region and central region is higher and the difference is relatively smaller. Through the KDE results of cities in these three regions, it can be found that the ULUE is lower in the regions with a higher proportion of the industry or pollution industry, and the improvement effect is smaller over time. We can find that the changes in ULUE of YREB from 2014 to 2019 show the characteristics of improved efficiency and narrowed gaps, whether it is an overall region or a local region, which further verifies the effectiveness of the policy.

### 3.2. Spatial Correlation Analysis

3.2.1. Global Spatial Autocorrelation Analysis

To analyze spatial correlation, global Moran's I was calculated by the Stata16.0 software platform. (Table 3). The results show that the global spatial evolution of ULUE from 2004 to 2019 showed the characteristics of "first decline and then increase". Except for 2006 and 2007, the global Moran index results are very significant. The results show that with the convenience of regional transportation and promotion of regional integration, the ULUE of the YREB has the characteristics of spatial dependence, and the spatial correlation has an upward trend, showing spatial agglomeration.

**Table 3.** Global spatial Moran's I of ULUE.

| Year | Moran's I | Z-Value | *p*-Value |
|------|-----------|---------|-----------|
| 2004 | 0.076 *** | 5.787 | 0.000 |
| 2005 | 0.055 *** | 4.466 | 0.000 |
| 2006 | 0.006 | 1.046 | 0.148 |
| 2007 | −0.008 | 0.092 | 0.463 |
| 2008 | 0.011 * | 1.439 | 0.075 |
| 2009 | 0.039 ** | 3.257 | 0.001 |
| 2010 | 0.011 * | 1.470 | 0.071 |
| 2011 | 0.031 * | 2.812 | 0.002 |
| 2012 | 0.012 * | 1.457 | 0.073 |
| 2013 | 0.062 *** | 4.835 | 0.000 |
| 2014 | 0.062 *** | 4.899 | 0.000 |
| 2015 | 0.058 *** | 4.759 | 0.000 |
| 2016 | 0.094 *** | 6.970 | 0.000 |
| 2017 | 0.086 *** | 6.556 | 0.000 |
| 2018 | 0.078 *** | 5.995 | 0.000 |
| 2019 | 0.126 *** | 9.171 | 0.000 |

Note: *, **, *** represent significance at the 10%, 5%, and 1% levels, respectively; the same below.

Additionally, the results of Moran's I is between [−1, 1], and its size reflects the strength of spatial autocorrelation. According to the results, we can find that the global autocorrelation of ULUE in the YREB is significant, reaching a minimum of 0.011 in 2008 and a peak of 0.126 in 2019. The increase in the Moran index indicates that the strategy of the YREB has regional synergy, and the impact on ULUE has the effect of regional linkage. Therefore, the study of ULUE in the YREB must consider the spatial relationship.

### 3.2.2. Local Spatial Autocorrelation Analysis

The global spatial autocorrelation can be analyzed by calculating the global Moran's I, but it cannot analyze agglomeration characteristics and spatial correlation of local cities. Analyzing the spatial correlation of local areas is conducive to the differentiation and investigation of the characteristics of different cities, and provides a reference for policies that adapt to local conditions. For the measurement of local spatial correlation, there are the local Moran scatter plot, Lisa agglomeration map, and hot-spot cold-spot map, etc. We use the Lisa agglomeration map and hot-spot cold-spot map to calculate the local correlation of ULUE. Therefore, based on the ArcGIS 10.7 software platform, we draw the Lisa map of ULUE in the YREB (Figure 5) and the distribution of hot-spot cold-spot map (Figure 6) to identify the agglomeration in local areas.

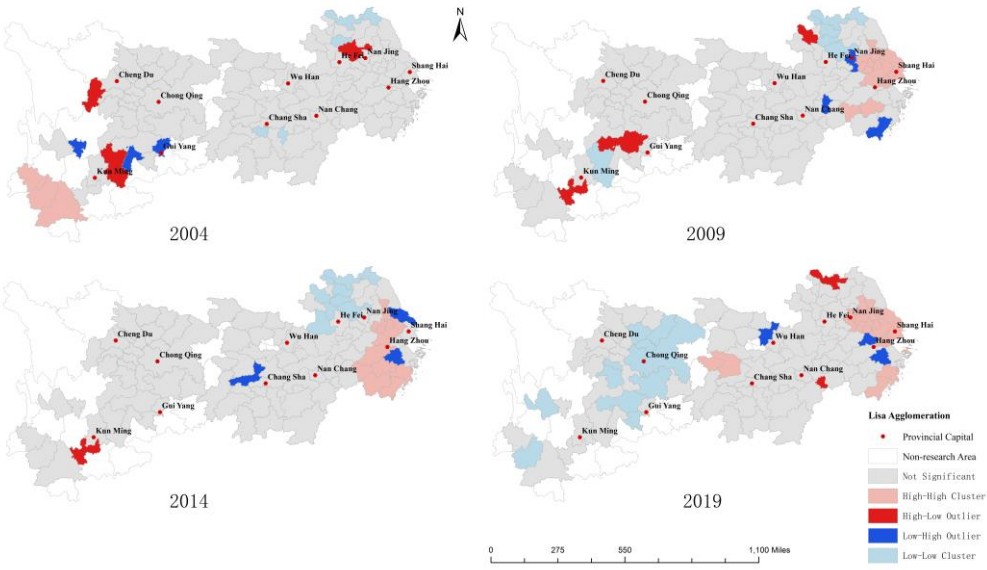

**Figure 5.** Lisa agglomeration index of ULUE.

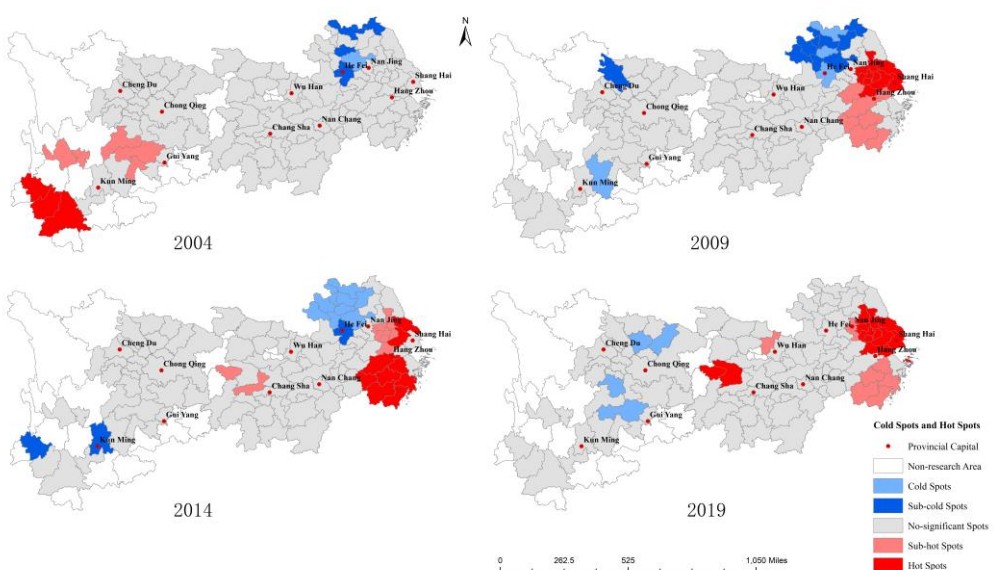

**Figure 6.** Cold spot and hot spot of ULUE.

Figure 5 shows the distribution of the Lisa agglomeration index in the YREB. With the evolution over time, the ULUE has shown significant differences among the three regions of YREB, with H-H agglomeration shifting to the cities in the eastern, L-L agglomeration shifting to the western cities in the region, and scattered distribution of L-H agglomeration

and H-L agglomeration. Specifically, from 2009 and later, the agglomeration characteristics of ULUE in local areas have changed significantly. Cities in the eastern region such as Nanjing and Shanghai have produced significant H-H agglomeration. Cities in the western region such as Chongqing and Zunyi have presented significant L-L agglomeration. Except for Zhangjiajie and Changde, which have H-H agglomeration characteristics in the cities of central region, other cities this region have not formed relatively obvious H-H or L-L agglomeration. The H-L agglomeration gradually shifted from the original Yunnan province to the eastern Anhui province, and the L-H ag-glomeration shifted from the original Anhui province to Yunnan and Sichuan provinces. From the situation of Lisa agglomeration, it can be seen that cities with high ULUE migrate to the eastern region, and cities with low efficiency migrate to the western region. At present, the development model of "low in the west but high in the east" has not changed.

Figure 6 reports the Getis-Ord Gi * index distribution of ULUE in the YREB, which is generally consistent with the Lisa agglomeration. The results show that the hot spot area shifts to the eastern area while the cold spot area shifts to the western area as time passes. Initially, cold spots appeared in Anhui Province, and hot spots appeared in Yunnan Province. In 2009, the hot spots in Yunnan disappeared, and there were many hot spots in Jiangsu, Zhejiang, and Shanghai. In 2014, the cold spots in cities in Anhui Province became weaker, the cold spots in Baoshan and Kunming became stronger, and second hot spots appeared in Zhangjiajie and Yiyang, and the hot spots in Jiangsu, Zhejiang and Shanghai became stronger. In 2019, the distribution characteristics of hot and cold spots were more obvious than in 2014. With the transfer of cold spots and hot spots, ULUE finally presents an overall pattern of the patchy distribution of hot spots in cities of the eastern region, scattered distribution of hot spots in cities of the central region, and patchy distribution of cold spots in cities of the western region. From the distribution results of the hot-spot and cold-spot map, we can find that there are still obvious differences between the east and west. Most of the cities enjoying agglomeration benefits are located in the eastern region, and there are more cold-spot regions in the western region.

### 3.3. Influencing Factors of Effects of ULUE

#### 3.3.1. Spatial Econometric Model Test

Due to the significant spatial correlation of ULUE, its influencing factors may also have a spatial spillover effect to ULUE. Therefore, we analyze the influencing factors and test the spatial spillover effect of ULUE based on the Stata16.0 software platform. However, we need to adapt three steps to select the spatial econometric model: LM test, Hausman test, and LR test (Table 4). The results of the LM test showed that the coefficients of Moran's I error term and lag term were all significantly positive, which indicates that it is necessary and feasible to use a spatial econometric model. Then, the Hausman test results significantly rejected the hypothesis of the random effect, which indicates that we should use the fixed effect rather than the random effect. The LR test shows that the Spatial Dubin model (SDM) is more credible and it should be employed, compared to the spatial error model (SEM) and the spatial autoregressive model (SAR).

**Table 4.** Test results of spatial econometric model.

| Indicator | Result |
|---|---|
| Moran's I | 12.010 *** |
| LMerror | 123.707 *** |
| R-LMerror | 6.066 ** |
| LMlag | 168.868 *** |
| R-LMlag | 51.227 *** |
| Hausman Test | 24.13 *** |
| LR_test (SAR) | 19.25 *** |
| LR_test (SEM) | 20.46 *** |

Note: **, *** represent significant at the 5%, and 1% levels, respectively.

### 3.3.2. SDM Results Analysis

To highlight the role of spatial models, we compare the results of OLS regression with the double fixed effect model with the result s of the SDM model with the adjacency weight matrix. The results can be seen in columns (1) and (2) of Table 5. At the same time, considering the robustness of the regression, we construct another two weight matrices named distance attenuation matrix and economic distance matrix, respectively. The results of the robustness are shown in column (3) and column (4). The results show that the OLS regression results are consistent with the unweighted results. However, the weighted coefficient significance and influence direction changed significantly, and the spatial lag coefficient indicates that there is a spatial spillover effect on the influence of urbanization level on ULUE. The above results further suggest that the use of a spatial effect model is necessary.

**Table 5.** Test results of spatial Dubin model.

| Indicator | OLS | Adjacency Matrix | Distance Attenuation Matrix | Economic Distance Matrix |
|---|---|---|---|---|
| Urban | −0.263 ** | −0.233 *** | −0.285 *** | −0.342 *** |
| | (−2.222) | (−4.680) | (−5.287) | (−5.825) |
| PD | 0.020 | 0.054 | 0.190 | 0.131 |
| | (0.114) | (0.280) | (0.950) | (0.639) |
| Gov | 0.060 | 0.020 | −0.004 | −0.036 |
| | (0.770) | (0.430) | (−0.089) | (−0.711) |
| IST | 0.057 | 0.016 | 0.028 | 0.035 |
| | (1.427) | (0.377) | (0.591) | (0.665) |
| LUS | −0.072 *** | −0.066 ** | −0.066 ** | −0.060 * |
| | (−2.930) | (−2.109) | (−2.053) | (−1.866) |
| Constant | 3.009 ** | | | |
| | (2.492) | | | |
| W* Urban | | 0.287 ** | 0.474 *** | 0.486 *** |
| | | (2.253) | (4.093) | (5.421) |
| W* PD | | −4.161 ** | −3.301 *** | −1.252 ** |
| | | (−2.027) | (−2.788) | (−2.235) |
| W* Gov | | 0.480 *** | 0.414 *** | 0.306 *** |
| | | (2.924) | (3.460) | (3.575) |
| W* IA | | 0.188 ** | 0.109 | 0.145 ** |
| | | (1.962) | (1.214) | (1.976) |
| W* IST | | −0.104 | −0.241 | −0.165 ** |
| | | (−0.452) | (−1.483) | (−1.979) |
| W* LUS | | 0.287 ** | 0.474 *** | 0.486 *** |
| | | (2.253) | (4.093) | (5.421) |
| $\rho$ | | 0.545 *** | 0.494 *** | 0.268 *** |
| | | (6.200) | (6.596) | (5.606) |
| $sigma^2$ | | 0.081 *** | 0.081 *** | 0.081 *** |
| | | (29.598) | (29.585) | (29.607) |
| LL | −260.715 | −295.666 | −286.729 | −289.926 |
| AIC | 563.431 | 615.332 | 597.457 | 603.852 |
| BIC | 679.638 | 681.009 | 663.134 | 669.528 |
| $R^2$ | 0.085 | 0.076 | 0.080 | 0.064 |
| Observations | 1870 | 1760 | 1760 | 1760 |

Note: *, **, *** represent significance at the 10%, 5%, and 1% levels, respectively, t values in parentheses.

Due to the result of the SDM model considering the spatial effect, it cannot directly explain the effect of influencing factors on the ULUE. To separate the effect of influencing factors, we further decompose the results of the SDM model as total, direct, and indirect effect, which is shown in Table 6.

**Table 6.** Decomposition effect results of spatial Dubin model.

| Indicator | Total Effect | Direct Effect | Indirect Effect |
|---|---|---|---|
| Urban | 0.382 ** | −0.277 *** | 0.659 *** |
| | (2.010) | (−5.053) | (3.336) |
| PD | −6.194 ** | 0.128 | −6.322 ** |
| | (−2.457) | (0.668) | (−2.516) |
| Gov | 0.805 *** | 0.007 | 0.798 *** |
| | (3.801) | (0.155) | (3.694) |
| IST | 0.273 ** | 0.031 | 0.242 |
| | (1.980) | (0.669) | (1.605) |
| LUS | −0.608 * | −0.070 ** | −0.539 * |
| | (−1.884) | (−2.279) | (−1.680) |

Note: *, **, *** represent significance at the 10%, 5%, and 1% levels, respectively.

The total effect of Urban is positive, the direct effect is negative, and the indirect effect is positive. With the transfer of the primary industry to the secondary and tertiary industries, the proportion of industry and manufacturing has gradually increased, bringing rapid economic development to the city. However, the increasing proportion of industry and manufacturing usually means increasing production pollution in the area. This unsustainable model will only lead to inefficient output if it has been developing over a long time. Additionally, the improvement of the local urbanization level will improve the industrial structure and technological development of the surrounding areas through economic exchanges, transportation and other means, which is beneficial to the ULUE of the surrounding cities. The total effect and indirect effect of PD are negative, but the direct effect is positive. Population concentration will lead to knowledge agglomeration and improvement in the innovation level, which can promote technological progress. However, the increase in population density will bring pressure on the local urban carrying capacity and siphon effect on the surrounding areas, which is not conducive to the urban labor structure and technological development of the surrounding areas, and finally show insignificant direct effects and significant negative effects. The total effect, direct effect, and indirect effect of Gov are positive. The level of government expenditure reflects the financial strength of the government and the activity of regional economic development. The higher the government expenditure, the higher the economy. The infrastructure and industrial development of developing countries need the economic input of the government. The higher the government's investment in economic construction, the easier it is to gather factors and produce innovation, which can improve the output efficiency of factor input. The total effect, direct effect, and indirect effect of IST are positive. The transformation of the industrial structure is the transformation from secondary industry to tertiary industry, from the pollution industry to the service industry. The transformation of the industrial structure reflects that the region pays attention to long-term development and moves towards a high-end intensive model, which is not only conducive to the ULUE of the region, but also affects the adjacent regions through the spatial spillover effect. The total effect, direct effect, and indirect effect of LUS are negative. The continuous increase in land use scale also reflects the relatively backward urban development and relatively low ULUE. Urban construction land continues to expand, which may be prone to the problems revitalizing assets, idleness, and low efficiency. Additionally, the continuous decrease in agricultural land causes agricultural risks, which is not beneficial to ULUE.

To further explore the heterogeneity of influencing factors on ULUE, we regrouped the samples based on the two characteristics of cities by resource type attribute and location, and we constructed the spatial weight and SDM again for regression (Table 7).

(1) The heterogeneity results of resource-based cities. "The National Sustainable Development Plan for Resource-based Cities (2013-2020)" divides China's cities into resource-based cities and non-resource-based cities. It points out that resource-based cities rely on natural resources. Promoting the sustainable development of these cities is an important way to achieve green economic development. For resource-based cities, due to their higher

dependence on resources, land use may be more inefficient, while non-resource-based cities will be more efficient. The results show that the impact of Urban on ULUE in non-resource-based cities is significantly positive while that in resource cities is negative, indicating that the consumption of resources as the main economic production mode is not conducive to the sustainable development of land. This phenomenon is also reflected in Gov's results. The government expenditure of non-resource-based cities is better. When the regional industrial structure is transformed, resource-based cities have a better protection effect on natural resources due to the transformation of traditional industries, showing a significant role in promoting ULUE. (2) The heterogeneity results of location, geographical heterogeneity. The absolute balance of regional development does not exist. The YREB contains cities in three regions, so there must be greater heterogeneity. The results of Urban show that the direct effect on ULUE gradually decreases from east to west, but the cities in the central region show a significant spillover effect. The government expenditure also showed consistent results, indicating that the eastern cities mainly engaged in manual manufacturing and paid more attention to improving ULUE. Because the unexpected output needs to be considered when calculating ULUE, the manufacturing industry in eastern cities has less unexpected output than that in western cities, so LUS shows a promoting effect on ULUE.

**Table 7.** Heterogeneity test results.

| Variable | Effect | Group: Resource-Based City | | Group: Position | | |
|---|---|---|---|---|---|---|
| | | YES | No | East | Central | West |
| Urban | Direct | −0.207 (−0.494) | 0.297 *** (2.772) | 0.460 *** (5.288) | −0.257 * (−1.750) | −0.838 (−0.974) |
| | Indirect | −11.074 *** (−2.849) | 6.150 * (1.700) | 2.632 (1.206) | 5.042 *** (3.303) | −13.258 (−1.049) |
| | Total | −11.281 *** (−2.838) | 6.447 * (1.752) | 3.092 (1.382) | 4.785 *** (3.059) | −14.095 (−1.083) |
| PD | Direct | 0.085 (0.791) | −0.069 *** (−3.368) | −0.025 (−1.140) | −0.192 *** (−5.334) | 0.150 (1.179) |
| | Indirect | 0.458 (1.506) | −0.199 (−0.667) | −0.269 (−0.902) | −0.248 ** (−2.123) | 0.636 (0.599) |
| | Total | 0.543 * (1.913) | −0.268 (−0.888) | −0.293 (−0.963) | −0.440 *** (−3.893) | 0.787 (0.723) |
| Gov | Direct | −0.045 (−0.433) | 0.054 *** (2.992) | 0.078 *** (3.556) | 0.012 (0.419) | 0.024 (0.200) |
| | Indirect | 0.392 * (1.735) | 0.489 *** (3.696) | 0.154 (0.951) | 0.190 *** (3.325) | 0.028 (0.104) |
| | Total | 0.347 * (1.768) | 0.544 *** (4.109) | 0.233 (1.417) | 0.201 *** (4.023) | 0.052 (0.218) |
| IST | Direct | −0.070 (−0.982) | −0.056 *** (−3.709) | −0.067 *** (−3.794) | −0.035 ** (−1.985) | −0.118 (−1.289) |
| | Indirect | 1.354 ** (2.240) | 0.250 (0.791) | 0.083 (0.217) | 0.294 (1.596) | 0.467 (0.768) |
| | Total | 1.285 ** (2.066) | 0.193 (0.601) | 0.017 (0.042) | 0.259 (1.370) | 0.349 (0.561) |
| LUS | Direct | −0.207 (−0.494) | 0.297 *** (2.772) | 0.460 *** (5.288) | −0.257 * (−1.750) | −0.838 (−0.974) |
| | Indirect | −11.074 *** (−2.849) | 6.150 * (1.700) | 2.632 (1.206) | 5.042 *** (3.303) | −13.258 (−1.049) |
| | Total | −11.281 *** (−2.838) | 6.447 * (1.752) | 3.092 (1.382) | 4.785 *** (3.059) | −14.095 (−1.083) |
| Other | R2 | 0.013 | 0.098 | 0.018 | 0.107 | 0.043 |
| | N | 688 | 1072 | 656 | 576 | 528 |
| | LL | −401.182 | 958.625 | 724.571 | 583.980 | −385.061 |
| | AIC | 826.364 | −1893.250 | −1425.142 | −1143.960 | 794.123 |
| | BIC | 880.770 | −1833.523 | −1371.308 | −1091.686 | 845.352 |

## 4. Discussion and Conclusions

### 4.1. Discussion

In 1949, China proposed promoting urbanization. After the reform and opening up in 1978, the promotion of urbanization entered a climax. The urban area continued to expand, and urbanization has achieved remarkable progress. In the past, to stimulate economic

growth, China implemented land reform, industrialization reform, and other events, which has raised its aggregate economic output from the 13th place in the world in 1949 to the second place in 2021. The Yangtze River Economic Belt is located in the south of China, with a pleasant climate. It gathers nearly 42% of the country's people with 21% of the land, which indicates that the efficient and sustainable use of land is extremely important. As an important supporting platform for China to build a high-quality development "growth pole," it provides resources and services for residents' life, employment, industry, service industry, manufacturing, and other industries, and is a rare research object. From the results of the spatial-temporal evolution of ULUE, we can find that there are still remarkable regional differences, which are related to the historical background of regional development. Most cities in the eastern region are located in plain or hilly areas, with good natural conditions and water transport conditions. In the early stage of economic development, China vigorously supported the development of cities in the eastern region to facilitate foreign exchanges and economic cooperation. Therefore, tertiary industry in cities in the eastern region was relatively developed while secondary industry in the central and western regions was relatively developed. When calculating the ULUE, we need to consider the unexpected output, so it reflects the differential development in different regions, which makes the cold spots and "L-L" clusters shift to the west, while the hot spots and "H-H" clusters shift to the east under the effect of spatial correlation. According to the hypothesis of "the first law of geography," the relevance between things will increase with the close proximity of geographical locations. With the circulation of talent and resources in various regions, technology spillover will enable better developed cities to drive the common improvement of ULUE in surrounding cities. However, the "growth pole theory" puts forward that balanced development between regions does not exist in reality. The "siphon effect" points out that the rapid development of economic activities will have a strong attraction to surrounding regions and a strong impact on their economies. Therefore, the "growth pole" of a region will bring double effects to the surrounding areas, which will not only aggravate the unbalanced development of the region through the gathering of resources, but also stimulate the increase in the overall economic scale. However, with the further development of the economy, the imbalance between the eastern and western regions is not conducive to high-quality development in the future. Therefore, it is necessary to play the role of regional synergy and coordinate regional common development. For example, the industrial upgrading and green development ability of surrounding areas can be promoted through the transfer of basic factor endowment and the collaborative use of technological innovation to promote the overall ULUE.

China's industrialization has three stages. The first stage was 1953–1957, in which priority was given to the development of heavy industry. The second stage was 1958–1978, which was a setback stage of socialist industrialization. The third stage, after 1978, is a new industrialization stage. With the continuous advancement of the industrialization stage, the goal of China's industrial development has gradually changed from economic priority to coordinated and sustainable economic and environmental development. However, due to the historical background of industrialization, the resource dependence and pollution level of large industrial provinces still need to be improved. For regions with relatively backward industrial development, the leapfrog development of the regional economy can be achieved by developing the industrial economy. In the past years, the level of industrial development can represent the development level and potential of the region, which has stimulated the efforts of various regions to support industrial construction. The continuous advancement of industrialization will have a negative impact on ecological resources, mineral resources, and the environment in the process of land development and utilization, which can only bring about short-term economic prosperity. From the spatial evolution and transfer of ULUE in the YREB, it can be found that the regions with high ULUE are dominated by light industry and comprehensive industry, while the central and western regions with low ULUE are dominated by industry or heavy industry, which reflects that the practice of taking economic benefits as the first goal in the past-production process is

not conducive to the improvement of ULUE. Additionally, although the dynamic evolution characteristics show that the urban gap is gradually narrowing, most cities are still in a relatively medium efficiency range, which needs to be improved as a whole. Therefore, the improvement of ULUE in the YREB in the future needs to go deep into the concept of green development, promote the reform and innovation of heavy industry and heavy pollution industry, require they produce under the premise of sustainable and green development, and improve ULUE by reducing unexpected output.

With the advance of population mobility and urbanization, the remote sensing data of night-light is more accurate to measure the level of urbanization. There are more than 110 cities in the YREB and with significant differences in urbanization development. China's overall planning goal for urban land is to make a reasonable layout of land on the premise of ensuring the sustainable development of the city. The overall planning goal for the city is to make an overall layout according to the local comprehensive characteristics. Land planning is the core of urban planning, and urban planning is the macro basis of land planning, which both include and restrict each other. The spatial effect of urbanization on ULUE shows that there are characteristics of high pollution and low return in the process of urbanization, which are more obvious in resource-based cities. For resource-consuming cities and highly industrialized cities, although there will be economic growth in the short term, when the resources in the region are exhausted, they will decline by double or even more times, and cause irreversible losses to the environment. Therefore, in the process of urbanization, the comprehensive strength of the region should be considered according to local conditions, such as the development of regional-characteristic industries, establishment of rural-characteristic industrial bases, etc., so as to give play to regional advantages and avoid blind urban expansion and waste of land.

*4.2. Conclusions*

Based on the Un_Super_SBM model, we measured the ULUE of YERB from 2004 to 2019 and used ArcGIS10.7 and Stata16.0 software platforms to analyze the spatial temporal evolution and spatial correlation of ULUE. We use the Un_Super_SBM model to calculate the ULUE of 110 cities in the YREB of China from 2004 to 2019. At the same time, the spatial-temporal evolution, spatial-temporal evolution, and spatial correlation of ULUE are analyzed. Additionally, the influencing factors of ULUE are explored. We draw four conclusions as follows:

(1) The time series characteristics show that the overall ULUE of the YREB is continuously improving. The ULUE of cities finally showed the characteristics of "lower in the west but higher in the east." The number of high ULUE cities in the YREB generally increase, but concentrate in the eastern region. The medium efficiency value cities concentrate in cities in the central region while most cities in the western region are still in low efficiency. The peaks of the KDE result in the whole region and sub-regions presented a "steep at first and then gentle" trend. The improvement in ULUE and regional synergy in these cities of the eastern and central regions is faster than cities in the western region.

(2) The spatial correlation of ULUE in the YREB has been increasing year after year, and the overall correlation is positive. The local spatial autocorrelation results show a spatial shift in ULUE. Specifically, the H-H agglomeration shifted to cities in the eastern region, the L-L agglomeration shifted to cities in the western region, and the L-H agglomeration and H-L agglomeration showed a scattered distribution. The Gi* index distribution results are consistent with the Lisa index results, and the hot spots and cold spots of ULUE are distributed regionally. Overall, the hot spots migrated to the east, and the cold spots migrated to the west, with a spreading trend.

(3) The results of the spatial Dobbin model show that Urban, Gov and IST can promote the improvement of ULUE, and PD and LUS can inhibit the improvement of ULUE. The decomposition effect shows that the direct effect of Urban is negative but the indirect is positive; the direct of PD is positive but the indirect effect is negative; both the direct

and indirect effects of Gov and IST are positive; both the direct and indirect effects of LUS are negative. The results of heterogeneity show that: economic activities in resource-based cities have a negative impact on ULUE, but industrial transformation will promote it. The economic activities of non-resource-based cities can promote ULUE, but the promotion effect of industrial transformation is not obvious.

**Author Contributions:** Conceptualization, L.Z., L.H. and K.D.; methodology, L.H.; software, L.H.; validation, L.Z., J.X. and L.H.; formal analysis, L.Z. and K.D.; investigation, J.X. and K.D.; resources, L.H.; data curation, L.Z.; writing—original draft preparation, L.H.; writing—review and editing, L.Z., J.X., L.H. and K.D.; visualization, L.H.; supervision, J.X. and K.D.; project administration, L.Z. and K.D.; funding acquisition, L.Z. and L.H. All authors have read and agreed to the published version of the manuscript.

**Funding:** This study is supported by the National Natural Science Foundation of China (grant number 72073054), Key Project of Jiangxi Provincial Social Science Fund (grant number 22ZXQH07), Nanchang Economic and Social Development Major Bidding Project (grant number ZDSK202202) and Chinese Government Scholarship Program (grant number CSC NO. 202209805005).

**Data Availability Statement:** Not applicable.

**Acknowledgments:** We thank the anonymous referees for their comments on the article. All of the errors and omissions remain our own.

**Conflicts of Interest:** The authors declare no conflict of interest.

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
