# Peer review of "Spatial-Temporal Evolution and Its Influencing Factors on Urban Land Use Efficiency in China’s Yangtze River Economic Belt"

_land, doi:10.3390/land12010076_

Round 1
Reviewer 1 Report
Based on the SBM model, this paper calculated the value of land use efficiency in the Yangtze River Economic Belt from 2004 to 2019, and analyzed its spatio-temporal evolution and spatial auto-correlation. It is of great significance to build SDM model and explore the driving factors that affect land use efficiency to improve land use efficiency, realize urban sustainable development and relieve urban land pressure. The article has a rigorous structure, detailed content and rich scientific research significance, but it still needs to be supplemented from the following aspects:
(1) It is suggested to add a description of the scientific issues in the abstract.
(2) It is suggested to reorganize the literature review. Through literature review, it is proposed that how to evaluate ULUE is the focus and difficulty of current research, which may be more reasonable. The summary of literature review and the proposal of scientific problems need to be combed again.
(3) In 2.1 Research area: it is suggested to add the data of land use in the Yangtze River Economic Belt, as well as economic and social data in the second paragraph.
(4) In 3.1.1 and 3.1.2, there are few reasons for the results, so it is recommended to add cause analysis.
(5) In the discussion part, the land use efficiency is discussed from three aspects: spatial heterogeneity, industrial economy and urbanization, which can be further deepened.
(6) There is too much content in the conclusion, so it is recommended to simply refine the conclusion.
Author Response
Thanks for the editor’s support and the reviewers’ comments concerning the manuscript. The reviewers’ comments are very helpful for us to improve our paper. We have revised the manuscript thoroughly, and the revised portion are marked in the manuscript using the “track changes” function in Microsoft Word (Please see the Revised Manuscript-Highlighted Version). In response to the reviewers’ comments and recommendations, the relevant amendments are summarized as follows (For your convenience, the reviewers’ original comments are in Italic, and our responses are in normal font):
[Comprehensive Comment]
Based on the SBM model, this paper calculated the value of land use efficiency in the Yangtze River Economic Belt from 2004 to 2019, and analyzed its spatio-temporal evolution and spatial auto-correlation. It is of great significance to build SDM model and explore the driving factors that affect land use efficiency to improve land use efficiency, realize urban sustainable development and relieve urban land pressure. The article has a rigorous structure, detailed content and rich scientific research significance, but it still needs to be supplemented from the following aspects:
[Comment 1] It is suggested to add a description of the scientific issues in the abstract.
[Response] Thanks very much for your comment. We have added a description of the scientific problems in the abstract. Please refer to the highlighted part of the Abstract (lines 12-15) for the modified contents.
The main purpose of this article is to measure the urban land use efficiency of the Yangtze River Economic Belt, and explore its evolution trend and influencing factors, so as to provide reference for policy formulation to promote efficient land use and sustainable development.
[Comment 2] It is suggested to reorganize the literature review. Through literature review, it is proposed that how to evaluate ULUE is the focus and difficulty of current research, which may be more reasonable. The summary of literature review and the proposal of scientific problems need to be combed again.
[Response] Thanks very much for your review. We have combed the existing literature of the article again, and systematically summarized the research results of this question, and proposed two marginal contribution values of this article. Please refer to the highlighted part of the introduction (lines 53-120) for the modified contents.
The marginal contribution of this paper has the following two points: First, the super efficiency SBM model (Un_Super_SBM) based on the unexpected output value measures the urban land use efficiency, which is improved on the basis of the traditional ULUE. Secondly, the changes and influencing factors of ULUE in the Yangtze River Economic Belt are discussed in depth, and the changes of ULUE are analyzed in detail through time series, spatial autocorrelation, spatial Dubin regression and heterogeneity test.
[Comment 3] In 2.1 Research area: it is suggested to add the data of land use in the Yangtze River Economic Belt, as well as economic and social data in the second paragraph.
[Response] Thanks for your comment. In the revised manuscript, We have added descriptions of economic and social data and land use data of the Yangtze River Economic Belt, and emphasized the significance of this study according to the characteristics of the data. It specifically includes the description of GDP, land loss, pollution and other data of the Yangtze River Economic Belt. Please refer to the highlighted part of the Materials and Methods (lines 154-170) for the modified contents.
According to the data of the statistical yearbook, the GDP of YREB in 2020 will exceed 47 trillion-yuan, accounting for more than 46% of the country. However, there is a big difference between the eastern region and the central and western regions. The GDP of the eastern region is 24.5 trillion-yuan, that of the central region is 11.1 trillion-yuan, and that of the western region is 11.6 trillion yuan. Although YREB's regional economic development is far ahead, it also faces many problems. Ac-cording to the data of “the Ecological Development Report of the Yangtze River Economic Belt (2019-2020)”, the water and soil loss has continued in recent years. In 2019, the area of water and soil loss will reach 293900 square kilometers, accounting for 20.14% of the land area. Although the pollutants and energy consumption of YREB will decrease in 2020, the situation is still not optimistic. In 2020, YREB's total wastewater discharge will account for 44.4% of the country's total, and the proportion of wastewater will exceed 40%. Green and efficient development needs urgent attention. Efficient and green land use is of great significance to sustainable economic development, and more attention should be paid to urban land with concentrated population distribution.
[Comment 4] In 3.1.1 and 3.1.2, there are few reasons for the results, so it is recommended to add cause analysis.
[Response] Thanks for your comment. Based on the analysis of the original results, we added the analysis of the causes of the results in 3.1.1 and 3.1.2. Please refer to the highlighted part of the Results (lines 294-303 & lines322-338) for the modified contents.
From the time series evolution results of land use efficiency in the Yangtze River Economic Belt and the three regions, it can be found that the overall level of land use efficiency is low, especially in the central and western regions. This is because we not only consider the effect of economic development, but also consider the pollution emissions in the production process when measuring the land use efficiency. In the past few years, the Yangtze River Economic Belt has gathered heavy industry and heavy pollution industry. Two of the four major industrial provinces are located in the western region and one in the central region. The distribution of high energy consuming industries and high pollution industries has a trend of "decreasing from east to west", so the land use efficiency is high in the east and low in the middle and west.
The evolution characteristics of temporal and spatial distribution clearly show the trend of temporal and spatial changes. Since the introduction of the planning policy of the YREB, the land use efficiency has been significantly improved. This is consistent with the result shown in Figure 2, the ULUE has changed significantly from 2014 to 2019. In 2019, high-efficiency regions concentrated in the eastern cities, middle efficiency concentrated in the central cities, and low efficiency concentrated in the western cities. Although the efficiency has been improved, the overall efficiency is not particularly high, because it is difficult to recover the resource consumption and ecological damage caused by the rapid economic development in the past in a short time. In addition, because the transformation of industrial equipment and production mode requires a long period, the overall land use efficiency is not high, and it is necessary to gradually promote the improvement of ULUE.
[Comment 5] In the discussion part, the land use efficiency is discussed from three aspects: spatial heterogeneity, industrial economy and urbanization, which can be further deepened.
[Response] Thanks for your comment. We further analyzed the above three aspects in the discussion department, including why regional coordination is important, the necessity of industrial industry and green development, and the internal mechanism of urbanization development. Please refer to the highlighted part of the Discussion and Conclusions (lines 552-562 & lines 577-586 & lines 600-606) for the modified contents.
According to the hypothesis of "the first law of geography", the relevance between things will increase with the close proximity of geographical locations. With the circulation of talents and resources in various regions, technology spillover will enable better developed cities to drive the common improvement of ULUE in surrounding cities. However, the "growth pole theory" puts forward that balanced development between regions does not exist in reality. The "siphon effect" points out that the rapid development of economic activities will have a strong attraction to surrounding regions and a strong impact on their economies. Therefore, the "growth pole" of a region will bring double effects to the surrounding areas, which will not only aggravate the unbalanced development of the region through the gathering of resources, but also stimulate the increase of the overall economic scale.
For regions with relatively backward industrial development, leapfrog development of regional economy can be achieved by developing industrial economy. In the past years, the level of industrial development can represent the development level and potential of the region, which has stimulated the efforts of various regions to support industrial construction. The continuous advancement of industrialization will have a negative impact on ecological resources, mineral resources and the environment in the process of land development and utilization, which can only bring about short-term economic prosperity.
China's overall planning goal for urban land is to make a reasonable layout of land on the premise of ensuring the sustainable development of the city. The overall planning goal for the city is to make an overall layout according to the local comprehensive characteristics. Land planning is the core of urban planning, and urban planning is the macro basis of land planning, which includes and restricts each other.
[Comment 6] There is too much content in the conclusion, so it is recommended to simply refine the conclusion.
[Response] Thanks for your comment. We have simplified the content, extracted three conclusions from the original four conclusions, and deleted the lengthy description, as shown in the highlighted text of the conclusion in the specific conclusion (lines 625 - 642).

Reviewer 2 Report
Improving urban land use efficiency is a feasible way to realize the sustainable development and alleviate the urban land pressure of the city. Authors calculated the value of urban land use efficiency in the Yangtze River economic belt, which is of practical significance. The research design of the article is scientific and reasonable, the research methods are properly selected, and the research conclusions are credible.
However, I suggest that the authors can further refine the innovation of the article and highlight the value and significance of the literature. At the same time, the discussion on the existing research results will be increased to enhance the dialogue with the existing literature.
After comprehensive judgment, I think this article meets the article standard of Land.
Author Response
Thanks for the editor’s support and the reviewers’ comments concerning the manuscript. The reviewers’ comments are very helpful for us to improve our paper. We have revised the manuscript thoroughly, and the revised portion are marked in the manuscript using the “track changes” function in Microsoft Word (Please see the Revised Manuscript-Highlighted Version). In response to the reviewers’ comments and recommendations, the relevant amendments are summarized as follows (For your convenience, the reviewers’ original comments are in Italic, and our responses are in normal font):
[Comprehensive Comment]
Improving urban land use efficiency is a feasible way to realize the sustainable development and alleviate the urban land pressure of the city. Authors calculated the value of urban land use efficiency in the Yangtze River economic belt, which is of practical significance. The research design of the article is scientific and reasonable, the research methods are properly selected, and the research conclusions are credible.
However, I suggest that the authors can further refine the innovation of the article and highlight the value and significance of the literature. At the same time, the discussion on the existing research results will be increased to enhance the dialogue with the existing literature.
After comprehensive judgment, I think this article meets the article standard of Land.
[Comment]
Improving urban land use efficiency is a feasible way to realize the sustainable development and alleviate the urban land pressure of the city. Authors calculated the value of urban land use efficiency in the Yangtze River economic belt, which is of practical significance. The research design of the article is scientific and reasonable, the research methods are properly selected, and the research conclusions are credible.
However, I suggest that the authors can further refine the innovation of the article and highlight the value and significance of the literature. At the same time, the discussion on the existing research results will be increased to enhance the dialogue with the existing literature.
After comprehensive judgment, I think this article meets the article standard of Land.
[Response]
Thanks very much for your review. We have combed the existing literature of the article again, and systematically summarized the research results of this question, and proposed two marginal contribution values of this article. Please refer to the highlighted part of the introduction (lines 53-120) for the modified contents. The revised contents are as follows:
At present, these researches on evaluation of ULUE mainly focuses on the necessity of ULUE measurement, the improvement of measurement methods and the promotion strategy of ULUE. First, most scholars believe that it is necessary to measure ULUE. Second, there are great differences in the methods of ULUE measurement in different researches. On the one hand, First, in terms of index system, there are large differences in the se-lection of different scholars. On the other hand, in terms of model construction, efficiency measurement includes parametric method, non-parametric method, multi-index dimensionality reduction and other methods. Third, there are multiple perspectives on the promotion strategy for ULUE. On the one hand, there are differences in the methods of evaluating ULUE changes. On the other hand, some scholars have analyzed the influencing factors of ULUE by constructing econometric models, and researched from the perspectives of industrial structure, population, and urbanization, respectively. However, from the current study, there still need further research from following two aspects: First, the existing literature is mostly evaluated from the ULUE of provincial cities or single prefecture level cities, and there are few studies involving the overall and local city in the YREB. Second, most of the existing models for measuring ULUE consider the unexpected output, but the efficiency value measured by the SBM model can only be between 0 and 1, which cannot analyze the effective decision-making unit, and it needs to be further improved. Based on the previous researches of ULUE, the marginal contribution of this article has two points: First, the super efficiency SBM model based on the unexpected output value (Un_Super_SBM) is used to measure the ULUE, which is improved on the basis of the traditional measurement of ULUE. Secondly, the changes and influencing factors of ULUE in the YREB are discussed in depth, and the changes of ULUE are analyzed in detail through time series, spatial autocorrelation, spatial Dubin regression and heterogeneity test.
